# Statistical Properties of Lasso-Shape Polymers and Their Implications for Complex Lasso Proteins Function

**DOI:** 10.3390/polym11040707

**Published:** 2019-04-17

**Authors:** Pawel Dabrowski-Tumanski, Bartosz Gren, Joanna I. Sulkowska

**Affiliations:** 1Centre of New Technologies, University of Warsaw, 02-097 Warsaw, Poland; p.dabrowski@cent.uw.edu.pl (P.D.-T.); b.gren@cent.uw.edu.pl (B.G.); 2Faculty of Chemistry, University of Warsaw, 02-093 Warsaw, Poland; 3Faculty of Physics, University of Warsaw, 02-093 Warsaw, Poland

**Keywords:** topology, lasso, proteins, threading, asphericity, prolateness

## Abstract

The shape and properties of closed loops depend on various topological factors. One of them is loop-threading, which is present in complex lasso proteins. In this work, we analyze the probability of loop-threading by the tail and its influence on the shape of the loop measured by the radius of gyration, distention, asphericity, and prolateness. In particular, we show that the probability of a trivial lasso for phantom polymer is non-zero even for an infinite structure, as well as that the threading flattens the loop by restricting its motion in one dimension. These results are further used to show that there are fewer non-trivial protein lassos than expected and select potentially functional complex lasso proteins.

## 1. Introduction

The physical background of polymer theory originates from classic works of Flory, de Gennes, Doi, Edwards, and others [1,2,3,4,5,6,7]. The tools they created enable the description of the behavior of linear or star polymers. However, they are unsuccessful in the description of circular chains. This stimulates intense debate and lively interest among the polymer community [8,9,10,11,12,13,14,15,16,17,18] especially as circular polymers arise naturally e.g., in bacterial or mitochondrial DNA [19,20,21]. Apart from the strictly topological effects present in circular polymers, such as knotting or linking [22,23,24,25,26,27,28], circular polymers may experience threading, which can largely influence the physical properties of such polymers [29,30,31,32,33,34,35,36,37,38,39]. To date, threading has been analyzed mostly for the rotaxanes, consisting of a macrocycle ring threaded by kinetically trapped linear molecules [40,41]. Only recently, lasso-like polymers composed of a loop with a threading tail attached, have been synthesized [42,43]. However, lasso-like threading was also discovered in proteins. Following [44] we call the threaded lasso topology the “non-trivial lasso” in opposition to non-threaded (therefore trivial) lasso.

The complex lasso proteins (also called pierced lasso bundles or tadpoles) consist of a loop formed by the main chain, closed by the covalent (e.g., disulfide) bridge, which is pierced by one or both of the tails (Figure 1A) [44,45]. Although previously unspotted, the whole class of complex lasso motifs is relatively common in proteins, with over 7000 chains identified so far, according to the LassoProt database [45]. Different piercing patterns were identified, including piercing “there and back” (the “L” class, up to 6 piercings), the chain winding around the loop (the “LS” class), or with both tails simultaneously piercing the surface (the “LL” class). Some motifs are presented schematically in Figure 1B. The lasso motif was proved to be functional in some cases, e.g., it regulates the function of the obesity-related protein—leptin [46,47], or acts as a plug for NTP uptake channel in case of amide-based lasso proteins—microcin or astexin [48,49,50,51,52,53]. The lasso peptides were also used to construct protein catenanes [54], or lasso peptide fusion proteins [55]. However, in general, the role of the lasso motif in protein biology is an open question.

To establish the function of a lasso, one could begin by analyzing the probability of the thread and its influence on the structure of the (bio)polymer. However, to date only the behavior of the ring with the static threading obstacles [32] and mutual threading of rings in a polymer melt [29] were analyzed, leaving the influence of the threading tail unexplored.

In this work, we bridge this knowledge gap. In particular, we analyze the probability of loop-threading and the influence of the thread on the loop size and shape. The shape of the loop is described by asphericity and prolateness of its corresponding ellipsoid of inertia. These results enable us to address the question of the lasso motif by comparing proteins with their polymeric counterparts. In particular, we show that there are fewer non-trivial lasso proteins than expected from polymer with equilateral loop [56] and tail modeled by equilateral random walk. We also select the protein lasso loops, whose structures are fairly different from those expected based on polymer simulation. We argue they can have a functional meaning.

## 2. Materials and Methods

### 2.1. Random Lassos Generation

Phantom lassos (polymers deprived of any interactions and volume) were created by connecting phantom loops and phantom tails. Phantom loops were created as equilateral polygons using the dedicated algorithm [56]. For *k*-gon, the algorithm samples uniformly k−3 dihedral angles and k−3 values from the set [−1,1] which are equal to differences between lengths of neighboring diagonals. From the valid data the polygon is recreated. Phantom tails were created as equilateral random walks starting from the first vertex of phantom loop with coordinates [0,0,0]. Three tests were done to validate the implementation of the loop sampling algorithm: acceptance rate test, curvature test, and HOMFLY-PT polynomial occurrence test (see Appendix A). Both loop length *N* and tail length *t* belonged to the set {10,20,30,40,50,60,70,80,90,100,120,140,160,180,200,250,300,350,400,500}, generating in total 400 pairs of every combination of loop and tail lengths. For each pair $10^{6}$ lassos were sampled. The whole process was implemented in Python 3.6 using Numpy 1.5 library.

### 2.2. Simulation Model

Our model consisted of a threaded loop (Figure 2A). The thread was infinite due to periodic boundary conditions to avoid loop slipping off of the thread. Polymers were modeled by adapting the coarse-grained Cα protein model [57] with protein-like bond interaction strengths and all equilibrium distance set to average Cα-Cα distance of $r_{0}=0.38$ nm. Neighboring beads interacted by a harmonic potential (with force constant equal $k=20,000\frac{\epsilon }{{\mathrm{nm}}^{2}}$). Other beads interacted by excluded volume effect described by the repulsive Van der Waals potential $4\epsilon {\left(\frac{\sigma }{r}\right)}^{12}$ had default value $4\epsilon \sigma^{12}=0.16777216\times 10^{-4}\left[\frac{{\mathrm{kJ}\mathrm{nm}}^{12}}{\mathrm{mol}}\right]$. This value matches VdW radius of 0.2 nm, slightly more than $2r_{0}$ to avoid polymers passing through each other. VdW potential was cut at a distance of 2 nm. No long-distance contacts or angular potential were added. The effect of the number of piercing tails was mimicked by making the radius of the thread beads w∈{0,1,2,3,4} times bigger than the loop beads. For the loop with N∈{3,4,5,6,7,8,10,12,15,20,30,40,50,70,100,150,200,300}$N_{B}=\max\left(\left\lfloor \frac{N}{2}+5\right\rfloor ,12\right)$ beads constituting the thread were used. As the presence of the thread forbids calculation of the loops that are too small, the length $N_{small}\left(w\right)$ of the smallest loops for the thread thickness *w* was equal $N_{small}\left(0\right)=3$, $N_{small}\left(1\right)=4$, $N_{small}\left(2\right)=6$, $N_{small}\left(3\right)=7$, $N_{small}\left(4\right)=7$. The system was encapsulated in periodic box of size $0.38\cdot N_{B}$ nm. To test different chain persistent lengths, simulations were carried out at temperatures T∈{10,50,100} in Gromacs units. $10^{9}$ steps were done for each condition (for a given temperature, loop length, and thread radius). To avoid biasing the results with the initial frame conformation, we removed the frames up to three times of time needed to reach 0.1 value of the chord correlation function [58]. The simulations were performed in Gromacs 4.6.5 [59]. 

### 2.3. Data Analysis

The lasso topology was determined using the *minimal surface* method [44] implemented in Topoly Python package [60]. The HOMFLY-PT calculation for validation of loop implementation was done using Ewing-Millett algorithm [61] implemented in Topoly package [60]. The radius of gyration and the lengths of axes of the ellipsoid of inertia were calculated using the g_gyrate program from Gromacs package. The fits were done using Gnuplot, and the obtained parameters are given up to 3 significant digits with the fitting errors in Appendix A. In our analysis we calculate the statistical probability (the fraction of structures with a given quantity among the set of whole structures), which we call simply “probability”.

### 2.4. Proteins Analyzed

The set of proteins analyzed was built by taking non-redundant representatives of all the protein structures present in February 2019 with a lasso (trivial or non-trivial), according to LassoProt (list in Appendix A). Clustering was done at 30% of sequence identity according to PDB implementation of the blastclust method.

### 2.5. Graphics

Lasso motif was visualized with a PyMOL plugin to identify lassos—PyLasso [62]. Molecular graphics were performed with UCSF Chimera [63], plots were created using matplotlib Python package or Gnuplot, and other figures were created using Pov Ray 3.6.

## 3. Results

### 3.1. Probability That a Lasso Is Complex

As stated in the introduction, the non-trivial lasso motif is relatively common in proteins. In fact, it is present in around 18% of all structures with disulfide bridges [44]. Therefore, we wanted to compare this number with the probability of non-trivial lassos in polymers. For this purpose, we generated a large ensemble of phantom lassos with various loop and tail lengths and determined their lasso motifs. Firstly, to get some intuition, we determined the probability of trivial lasso for the chain with equal lengths of loop and tail (Figure 3A). As can be seen, probability tends towards a value $P_{\infty }\left(L_{0}\right)=0.19$, which indicates that even an infinite lasso can be trivial (i.e., unthreaded) with non-zero probability. This follows from the loop occupying different regions of space than the tail, which is possible, as the space itself is infinite.

The probability of forming a trivial lasso should depend on both the tail and loop lengths. However, the dependence we obtained cannot be fitted with a simple sum of two exponential decays. To investigate this phenomenon further, we calculated the probability of trivial lasso for a fixed loop (Figure 3B) and tail (Figure 3C) lengths. Comparison of both cases shows that the elongation of the loop has a more sound effect on the probability of a trivial lasso. This may be attributed to the fact that the area of the surface increases faster than the length of the loop surrounding it. The area of a surface bound by ring polymers in melt without excluded volume was estimated to scale with the number of monomers in the boundary as $N^{1.03}$ [29]. Calculating the same dependence using protein data in LassoProt gives the power law $N^{1.19}$ (see Appendix A). The decay of trivial lasso probability for both fixed loop and tail length can be fitted with a sum of exponents with a free parameter (values in Appendix A):(1)$$P\left(N;L_{0}\right)=P_{\infty }\left(L_{0}\right)+c_{\alpha }\exp\left(-\alpha N\right)+c_{\beta }\exp\left(-\beta N\right)$$

As one characteristic length is always an order of magnitude smaller than the other one, here we observe short- and long-chain regimes. This seemingly contradicts earlier results on piercing obtained for the polymeric catenanes, where simple exponential decay was reported. However, these were either asymptotic cases, when the influence of the small length scale is absent [64,65], polymer melts [66,67] where many rings can thread each other, or the models with excluded volume [22,23]. The latter difference especially indicates the nature of our two-regime observation. With no excluded volume and no restriction on angles, there is nothing to prevent the polymeric tail from making a sharp turn back towards the surface, or pierce the surface very close to its boundary. This effect is, however, expected to fade upon an introduction of non-zero polymer thickness, leaving only the single exponential decay, consistent with the previous results. Therefore, the probability of the trivial phantom lasso can be described as a surface parametrized by both tail and loop lengths as:(2)$$P\left(N,t;L_{0}\right)=P_{\infty }\left(L_{0}\right)+c_{\alpha ;t}\exp\left(-\alpha_{t}t\right)+c_{\alpha ;N}\exp\left(-\alpha_{N}N\right)+c_{\beta ;t}\exp\left(-\beta_{t}t\right)+c_{\beta ;N}\exp\left(-\beta_{N}N\right)$$

The fitted surface is shown in Figure 3D. In particular, this form of the probability dependence shows why we could not fit a sum of two exponents in case of an equal loop and tail length.

For a non-trivial lasso, it is interesting what type of threading pattern may be observed. We analyzed the total number of piercings (independently of their direction i.e., merging *L* and LS classes) for the case of an equal loop and tail length (Figure 4A). For all analyzed conditions, the most probable non-trivial lasso is the singly pierced $L_{1}$ motif. From the beginning, probability of the $L_{1}$ motif decreases, reaching a plateau for very large chains. We note that it is different behavior than in the case of e.g., knots, where the probability of the simplest structures drops nearly to 0 when more complex knots occur [68,69,70,71]. The non-vanishing of the simplest lassos in the asymptotic case may be explained by considering the parts of the tail delimited by the piercings separately. After performing the first piercing, there is non-zero probability $p_{\mathrm{drift}}$ that the chain would go away from the loop. Because the tail is modelled as a memoryless random walk, the same reasoning as in case of the trivial asymptotic lasso applies, and therefore the probability $p_{\mathrm{drift}}$ should be similar to the probability of infinite trivial lasso $P_{\infty }\left(L_{0}\right)$ (but not necessarily equal).

In fact, this reasoning may be iterated in the case of doubly, triply, quadruply etc. threaded loop. As a result, the asymptotic probabilities $P_{\infty }\left(L_{n}\right)$ of obtaining a non-trivial lasso with *n* piercings is expected to form a decreasing series. Indeed, such a situation is visible in Figure 4A, where the traces for different total piercing numbers stabilize for large structures in the decreasing order of the piercing number. This distribution may be described in terms of the standard probability theory. In fact, when analyzing the probability of several piercings, one could consider the tail drifting apart as a “success”, and tail piercing as a “failure”. This leads to a geometric distribution, with the “success” parameter equal to the probability of infinite trivial lasso $P_{\infty }\left(L_{0}\right)$. In fact, the values at which the probability traces stabilize in Figure 4A are lower than in such framework indicating that the geometric distribution is too idealized description of the asymptotic case.

Apart from studying the asymptotic case, one could analyze how the distribution of lasso non-trivial motifs depends on the lasso size. In particular, as can be seen in Figure 4A, for lassos with N=250 beads in each of the loop and the tail, the spectrum is dominated by structures with at least 5 piercings (probability of “other” >0.5). In fact, this cutoff number of piercings may be used to characterize the spectrum of non-trivial lasso motifs for a given length *N*. In particular, we can define the domination number *d* as the maximal number of piercings *p*, for which a non-trivial lasso with at least *p* piercings dominates the spectrum (e.g., d(250,250)=5), that is:(3)$$d\left(N,t\right)=\max\left(p\right):P_{N;t}\left(n\geq p\right)>1/2$$with $P_{N;t}\left(n\right)$ being the probability that a lasso with the loop length *N* and tail length *t* has n>0 piercings. From the plot of the domination number *d* as a function of the lasso length *N* (Figure 4B) one can see that although $L_{1}$ is the most common lasso motif, even for relatively short structures (N=30 beads) it is more probable to generate more complicated lassos than $L_{1}$. On the other hand, the value d(N,t) is bounded from above, as for some number of piercings $p_{0}$ the probability $P_{N;t}\left(n\geq p_{0}\right)$ cannot be larger than 1/2, as it is the tail of the convergent series. Indeed, we see a stabilization indicating that sampling lassos with more than 500 beads in the loop and tail would not improve much our knowledge about the distribution of non-trivial lassos for those lasso sizes.

### 3.2. Shape Parameters of a Lasso Loop

Apart from the threading probability, another question concerns the influence of the threading on the shape of the loop. To analyze the dependence of the shape of the loop on the threading type, we generated the ensemble of structures from the simulation of unconstrained movement of a coarse-grained loop. To obtain the threading effect, the loop was pierced by a single thread with the radius of the thread imitating the piercing pattern (see Figure 2A and Materials and Methods).

In each simulation, the size of a loop was described by the radius of gyration $R_{g}$ and distention *D* (Equation (Equation 6)), and its shape by asphericity *A* and prolateness *P* of the ellipsoid of inertia of the loop (Figure 2). The description by an ellipsoid of inertia has a long tradition [72] and was used to analyze topologically trivial [73,74,75,76] as well as knotted [68,77] polymers. Moreover, the analytical results in the limit of an infinitely flexible chain are known [78,79,80]. All the analysis included the results obtained at the highest temperature analyzed, at which the chain was the most flexible. However, the results at the lower temperatures were qualitatively equivalent (see Appendix A).

The radius of gyration measures the average size of the polymer chain:(4)$$R_{g}=\sqrt{\frac{1}{N}\sum_{i=1}^{N}r_{i}^{2}}$$with $r_{i}$ denoting the position of the loop beads. The mean radius of gyration as a function of the loop length is shown in Figure 5A. The largest differences in the $R_{g}$ can be seen for large loops, where the behavior of the free (non-threaded) loop stands out from the threaded cases. On the other hand, the traces for threaded loops are parallel for long chains, indicating that for sufficiently long chains the thread thickness does not play a role.

To quantify the effect of the thread, the data were fitted with the standard formula [36]:(5)$$<R_{g}>=c_{r}+a_{r}N^{\nu }$$

To exclude the effect of small, rigid loops, we fitted the data for loops with at least 30 beads. The result for the non-threaded loop (ν=0.582) is close to the exponent estimated for the self-avoiding walk (ν=0.5874±0.0002 [81]), and the difference may stem from the non-fixed length of the bonds. In fact, all the obtained values of the scaling exponent were similar independently on the thread thickness, with an average value $\bar{\nu }=0.580$ (see Appendix A). This can be explained, as the existence of a thread does not change the solvent properties, nor the dimension of the space—the factors which according to standard Flory theory determine the power law. Moreover, it does not force the loop to squeeze (like the presence of knot does). In fact, the scaling of radius of gyration was shown to be independent on some topological factors, such as cyclization of the chain [36]. Therefore, following [36], to obtain a better fit of other parameters, we fitted the function with fixed the value of scaling exponent ν=0.59. The results of the fit are contained in Table 1.

Despite no effect on the scaling, the thread does influence the preexponential factor $a_{r}$. For the same scaling exponent, the ratios of parameters $a_{r}$ describe the ratio of the values of radius of gyration in different cases (for sufficiently long polymer). The ratios of $a_{r}$ compared to the unthreaded case is also given in Table 1. The analysis of the ratios shows that the threaded loops with the same length have slightly larger $R_{g}$, which results from spreading the chain in two dimensions perpendicular to the thread. On the other hand, as expected, the thread thickness does not influence the value of $R_{g}$ asymptotically (the discrepancies in $a_{r}$ parameter are smaller than our confidence interval).

The radius of gyration is a good measure for large loops, where the scaling behavior can be analyzed. However, for small loops, the excluded volume effects constraint the movement of the chain, flattening it, independently of the presence of the threading. Therefore, we defined a new quantity, called distension *D*, which compares the $R_{g}$ with the value for a flat, regular polygon with the same number of beads:(6)$$D=\frac{2R_{g}}{r_{0}}\sin\left(\frac{\pi }{N}\right)$$

The maximally stretched loop, symmetrical, with relaxed bonds, has distention value of 1. Higher values can be reached if loop bonds are stretched beyond equilibrium value $r_{0}=0.38$ nm or lower if it is contracted. The plot of distension is shown in Figure 5B. In particular, it shows that the presence of the thread elevates the distension, again indicating the flattening of the loop. Similarly to $R_{g}$, the effect of the presence of the thread is visible also for large loops, where the effect of the thread thickness vanishes.

The flattening and elongation are well visible in the plots of asphericity and prolateness. The asphericity measures the deviation from the spherical shape. In terms of the ellipsoid of inertia semi-axes *a*, *b*, and *c* it is given by:(7)$$A=\frac{{\left(a-b\right)}^{2}+{\left(b-c\right)}^{2}+{\left(c-a\right)}^{2}}{2{\left(a+b+c\right)}^{2}}$$(we used the definition of Rawdon et al [68], as in this definition the asphericity is an unbiased parameter). The asphericity attains values in the interval A∈[0,1], with 0 for a sphere, 0.25 for a flat disk and 1 for a stick (see also Figure 2C). In our case, the excluded volume forces the small loops to attain more planar shapes (as seen also by the distension), which results in the values of asphericity characteristic for disk-like structures. For non-threaded loop, the asphericity has a very shallow minimum for a size of 7 beads (similar shapes were also registered for loops without excluded volume [68]). However, the presence of the thread modifies this behavior. The traces for threaded loops do not have a minimum and therefore approach their asymptotic values from above.

To calculate the effect of thread thickness on the asymptotic value of asphericity, we fitted the asphericity of all threaded loops larger than 30 beads with the equation [74]:(8)$$<A>=A_{\infty }+a_{A}N^{\mu }$$

The results of the fit are contained in Table 1. For a non-threaded case, we observe the value $A_{\infty }=0.0708$, comparable with earlier results ($A_{\infty }=0.078$ in [77]). The presence of the thread increases the asymptotic value of the asphericity to the average value ${\bar{A}}_{\infty }=0.589$ with no visible dependence on the thread thickness. However, the thread thickness manifests itself by the rate by which the limit value is obtained. This rate of approaching the limit may be quantified by the scaling exponent μ, visible as the curvature of the plot.

The second shape parameter used is the prolateness, also referred to as the nature of asphericity. It can be calculated for the aspherical loops according to the equation [68]:(9)$$P=\frac{\left(2a-b-c)(2b-c-a)(2c-a-b\right)}{2{\left(a^{2}+b^{2}+c^{2}-ab-bc-ca\right)}^{\frac{3}{2}}}.$$

The results of the fit are contained in Table 1. Prolateness attains its values in interval P∈[−1,1], with negative values attained for oblate ($\frac{a+c}{2}<b$, M&Ms-like shape) and positive for prolate structures ($\frac{a+c}{2}>b$, rugby-ball-like shape, assuming a≥b≥c, see also Figure 2C).The prolateness dependence on the threading is shown in Figure 5D. All small, threaded loops are oblate, becoming more elongated in one direction (more prolate) with the extension of the chain. The thickness of the thread has a smaller effect on the prolateness than just the thread existence.

To quantify this effect, we turned towards scaling of the prolateness. However, there is an inconsistency in the literature according to the scaling law with either linear [74], or almost inverse square root power law [79] scaling for intermediate-length chains. As in our case, the traces are concave, and we could not fit the data with variable exponent, we turned towards the almost inverse square root dependence:(10)$$<P>=P_{\infty }+a_{P}N^{-0.47}$$

The asymptotic prolateness for non-threaded loop ($P_{\infty }=0.331$) agrees with previous results [68,77]. The presence of the thread increases the prolateness, indicating that the threaded loop is more elongated in one dimension. However, conversely to asphericity, the asymptotic values of the prolateness still seem to be affected by the thread thickness. This effect is visible both in the plot as in the fitting parameters. As we expect the prolateness to be asymptotically free of the thickness of the thread, this indicates that used scaling law is valid only in a narrow interval of chain lengths and better scaling should be used.

The difference between the threaded and non-threaded case is well visible especially by plotting the traces in the space of the ellipsoid semi-axis, along with rising loop length (Figure 5E). We used this presentation, rather than the asphericity-prolateness space used in other works, as in the latter, part of the space is forbidden [74]. As the very small loops should be treated separately, the plot encompasses the traces starting from loops with at least 8 beads. For 8 beads, the presence of the thread still flattens the loop, increasing the asphericity and decreasing the prolateness. This shifts the starting point towards the right-bottom corner of the plot compared to the non-threaded case. The difference in approaching the asymptotic asphericity value is visible as a different sign of the curvature of the traces. The threaded loops tend towards the same point (asymptotical independence on the thickness of the thread), which is, however, distinct from the limit point for the unthreaded loop (the dependence of the shape on the thread existence).

### 3.3. Comparison Of Simulated Polymers with Complex Lasso Proteins

One of the biological realizations of the lasso geometry are the complex lasso proteins. As stated in the introduction, the general role of the lasso motif in proteins is still unknown; however, in some cases (miniproteins, leptin) the motif was shown to be functional. Our results indicate that the lasso may be functional in many other cases. In particular, similarly as in the case of knots [82], we analyzed the probability of a protein trivial lasso versus the probability of trivial lasso occurrence in phantom lasso polymers (Figure 6A). In most cases the points corresponding to the protein data locate above the surface describing polymers, indicating that the non-trivial lassos are scarcer in biopolymers. However, some particular points lie well below the polymer surface. This may be regarded as a hallmark of the importance of the lasso structure, as the loop or the tail has to be additionally stabilized by some enthalpic effects despite entropic tendency to form a trivial structure.

On the other hand, the number of non-trivial lassos in proteins (18% of all proteins with disulfide-based loops [44]) can be compared with the expected number of non-trivial structures in our polymeric model. Therefore, we estimated the threading probability for each chain, using Equation (Equation 2). As in general the protein lassos possess two tails, and some protein chains possess more than one lasso loop, we calculated the probability of at least one threading, using the inclusion-exclusion principle (see Appendix A for details). Next, we calculated the expected number of chains with non-trivial lasso structure as a function of chain length. The result of the calculation is shown in (Figure 6B). Although the curves are highly variable, the trend shows that the observed number of non-trivial lassos is smaller than the expected number, especially for long chains (more than 200 residues).

To identify the protein structures, for which the non-trivial lasso may be functional, we selected the non-trivial structures, with very low threading probability (<0.2) estimated for the polymer model, reasoning as previously that either the loop or the thread has to be additionally stabilized. In the probability estimation, we also included the effect of multiple piercings through the loop, by treating the piercing number as a geometric distribution, as argued earlier.

The set of structures with potentially functional lassos selected this way included all miniproteins (18 structures with the average threading probability equal to 0.024), for which the non-trivial lasso motif was already shown to be functional. Moreover, the set contained a few other small toxic, antimicrobial, defensin-like or immune system related proteins (27 structures with the average threading probability equal to 0.122) with the same lasso motif as miniproteins (see e.g., Figure 1A). Apart from those, we identified 8 hydrolases or hydrolase inhibitors (with the average threading probability equal 0.141), one sugar binding protein and the proteins with 6 piercings (metal-binding protein with PDB code 6bdj and oxidoreductases with PDB codes 4qi7 and 5vg2). The full list of identified proteins with their estimated threading probability and the spectrum of all threading probabilities are contained in Appendix A. Apart from those, the chemokines and other signaling proteins with $L_{2}$ motif were pointed out previously based on the simple statistical analysis [44]. In fact, most of these proteins have the piercing probability just above 0.2 according to our estimate.

On the other hand, one could select the structures for which the threaded loop has an unexpected shape measured by the values of the asphericity or prolateness. In particular, we showed and argued for the piercing results in the flattening of the loop, implying the increase of asphericity, compared to the non-threaded loop of the same length. Therefore, as the threaded loops are larger in general (Figure 6C) than non-threaded, one would expect the shift of the asphericity distribution towards higher asphericity values, upon threading. However, the entirely opposite effect is seen in proteins, where the threaded loops are more spherical (Figure 6D). Again, we associate this counterintuitive effect with possible stabilization of the loop or the tail balancing the natural entropic tendency to increase the asphericity. However, the results based on shape parameters are much vaguer, and more subtle tests, taking into account also internal protein geometric features are needed.

## 4. Discussion

In this work, we analyzed the probability of piercing and its result on the shape of the loop for a broad range of loop and tail lengths. In contrast with previous works, in our case, the loop was threaded by a tail with one free terminus. In particular, we showed that the probability of trivial phantom lasso is non-zero even for the infinite structure. This has a direct influence on the distribution of non-trivial lassos, with the simplest $L_{1}$ type being the most common pierced lasso regardless of the loop and the tail lengths. On the other hand, we showed that the threading flattens the loop, which is visible in both asphericity and prolateness. The effect of the thread is visible for all sizes of lassos; however, the influence of the thickness of the thread asymptotically disappears.

The shape parameters were calculated for the loop with excluded volume. The excluded volume affects in particular the shape of the small loops. It would, therefore, be interesting to see how the asphericity and prolateness of the loop depend on the loop thickness. The excluded volume may also have an important effect on the probability of non-trivial lasso. In this work, we analyzed the probability that lasso is non-trivial for phantom chains, also registering an effect specific to small structures, increasing this probability. We attribute this effect to sharp turns of the tail or loop, which would disappear upon introduction of non-zero chain thickness.

Apart from the excluded volume, the probability of threading and its influence on the shape of the loop may be modulated by other factors, such as chain persistence length, confinement, or by imposing some specific distribution of planar and dihedral angles. For unthreaded loops, some works concerning the influence of these parameters on the shape of the loop have already been done [73,74,83]. In fact, confinement was shown previously to increase the probability of knotting, both in the case of polymers and in proteins [25,84,85,86,87]. In the case of lassos, confinement is expected to decrease the asymptotical probability of a trivial lasso, and therefore change the distribution of total piercing number.

Such modifications would enable adjusting the model to the specific case, e.g., prediction of lassos in proteins. Based on the general results, we singled out some protein structures for which the lasso motif may be functional. In accordance with previous studies, our analysis selected in particular the antimicrobial proteins, with single threading. The structural and functional similarity to miniproteins indicates that those proteins may also act as plugs for some crucial channels. This, in turn, creates the possibility of medical use for small complex lasso proteins. Most of the other proteins selected by our analysis are enzymes. This resembles the case of proteins with knotted backbone, around 80% of which are enzymes [88]. For those proteins, the existence of knot-induced rigidity was shown to create places favorable for the enzymatic active sites [89]. Possibly, a similar effect is present in the case of complex lasso proteins, with the rigidity induced by the covalent loop. Enhancing the model with excluded volume mimicking cell crowding or imposing the protein-like distribution of the planar and dihedral angles could tune the predicting power of the analysis to indicate more subtle examples of lassos, for which the threading must be additionally stabilized, and therefore is suspected to be important for the protein structure or function. Similarly, such enhanced models could allow the obtaining of more detailed statistics on the expected number of proteins with complex lasso topology. However, as the static polymeric approach does not include the effect of protein folding, we expect such results to overestimate the number of complex lasso proteins. This also shows the necessity to investigate further the topic of folding of such structures.

Another way to test the stabilization of the tail is the direct measurement by experimental or simulated pulling. In general, the analysis of the thermodynamic properties of lasso (bio)polymers, including their dynamics, and formation constitutes the natural continuation of this work. Until now, only dynamic studies of mutual loop-threading and study on the thermal unthreading of one lasso protein have been performed [13,90].

Threading is a fascinating topological phenomenon, which, although it has profound effects on the shape and dynamics (topological glue) of polymers, has not yet been studied extensively. With this work and our parallel of the complex lasso proteins in mind, we hope, however, to positively stimulate the development of the study of the threading effects on the polymers with its applications in biology.

## Figures and Tables

**Figure 1 polymers-11-00707-f001:**
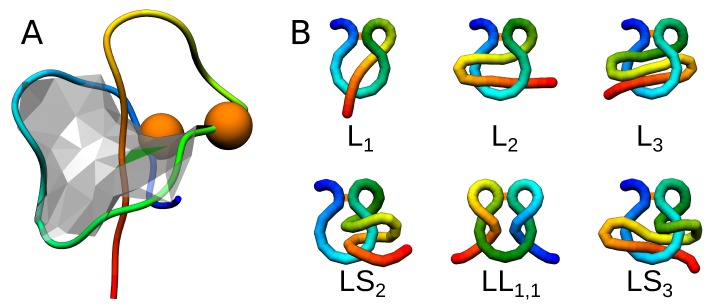
Complex lasso proteins.(**A**) An exemplary complex lasso protein (PDB code 2mxq) with one piercing through the loop ($L_{1}$ type). The orange beads denote the cysteine residues; (**B**) Schematic depiction of six complex lasso motifs, with the orange bar denoting the bridge.

**Figure 2 polymers-11-00707-f002:**
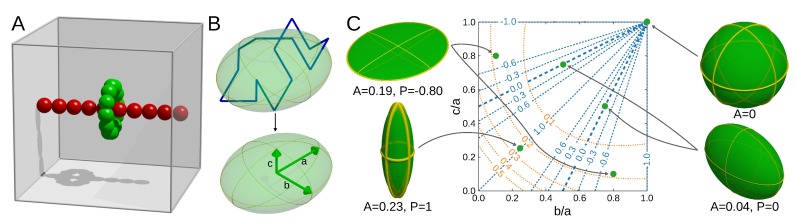
Description of the model and shape parameters. (**A**) The schematic presentation of the model used in the simulation–the shape of the freely moving green loop was analyzed; (**B**) The polymer with its ellipsoid of inertia (top) and the semi-axis of the ellipsoid used to calculate the shape parameters (bottom); (**C**) The exemplary ellipsoids with their location on the contour plot of the asphericity (orange curves) and prolateness (blue curves) in the space of the fraction of ellipsoid semi-axes a>b>c. Note that for spherical ellipsoid (b/a=c/a=1), the prolateness is undetermined.

**Figure 3 polymers-11-00707-f003:**
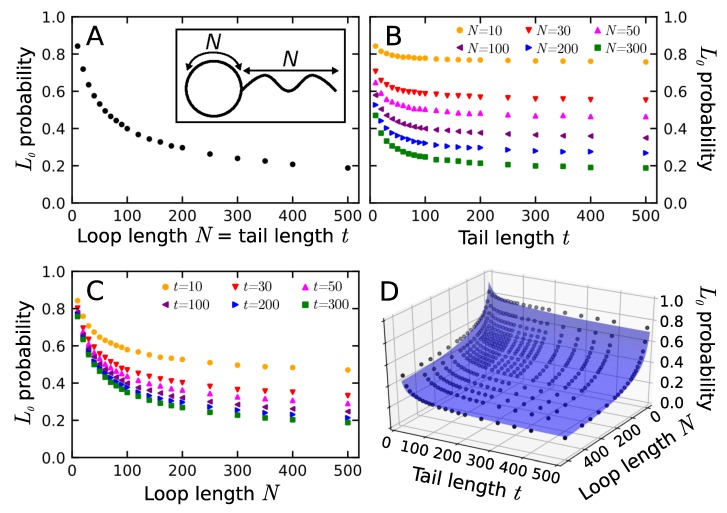
The probability of trivial lassos. (**A**) A plot of the probability for an equal length of the tail *t* and the loop *N*. In the inset, the schematic structure of trivial lasso; (**B**) Probability of trivial lasso vs tail length *t* for fixed loop length *N*; (**C**) Probability of trivial lasso vs loop length *N* for fixed tail length *t*; (**D**) The surface of trivial lasso probability in the space of loop and tail length. In the case of (**B**) and (**C**) only selected traces were plotted to maintain transparency. For all the data see Appendix A.

**Figure 4 polymers-11-00707-f004:**
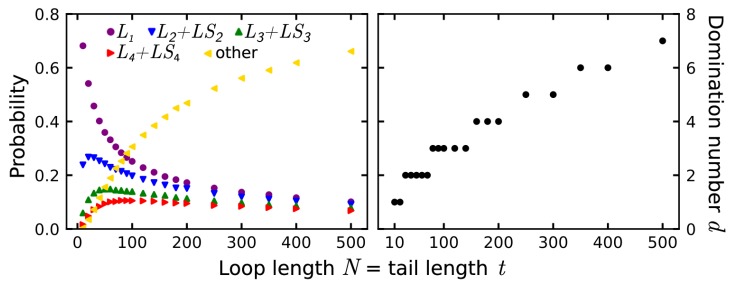
Analysis of non-trivial lassos. (**Left panel**) The probability for a given number of piercings as a function of the loop size for an equal loop size *N* and tail length *t* for non-trivial lassos only. “Other” means lassos with ≥ 5 piercings.; (**Right panel**) The domination number *d* as a function of the loop length *N* and tail length t=N for non-trivial lassos. The domination number *d* is a maximal number of piercings *p* for which lassos with ≥p piercings are the majority of non-trivial lassos with given *N* and *t* (Equation (Equation 3)).

**Figure 5 polymers-11-00707-f005:**
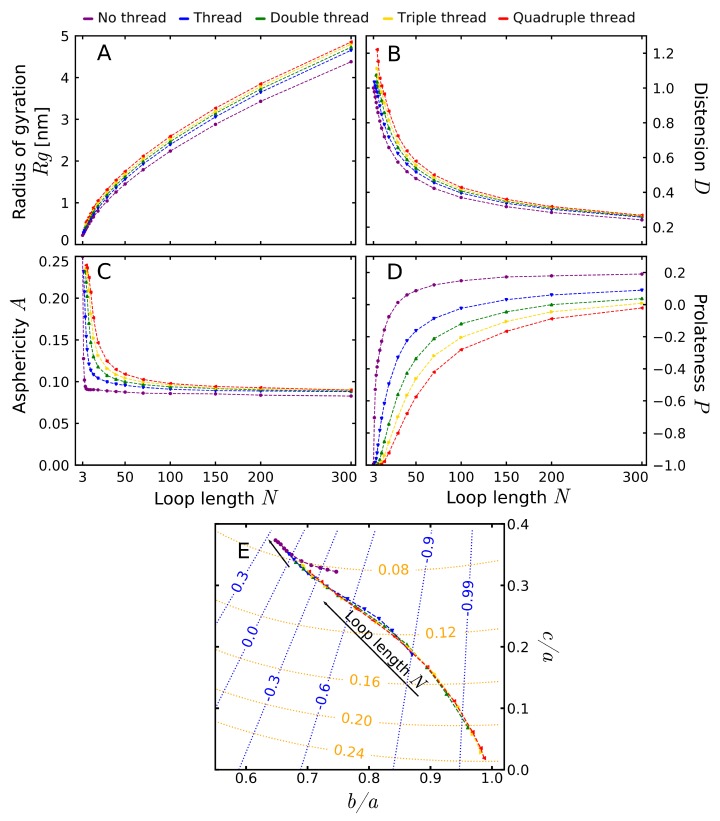
The dependence of shape parameters on the loop length and thread thickness. (**A**) Radius of gyration; (**B**) Distension; (**C**) Asphericity; (**D**) Prolateness; (**E**) The trajectories of the loop shapes in the space of the fraction of ellipsoid semi-axes a>b>c. The traces start from the loops with at least 8 beads. The contour lines of asphericity (orange) and prolateness (blue) were added. Note that the thread thickness determines the smallest possible loop (for details see Materials and Methods).

**Figure 6 polymers-11-00707-f006:**
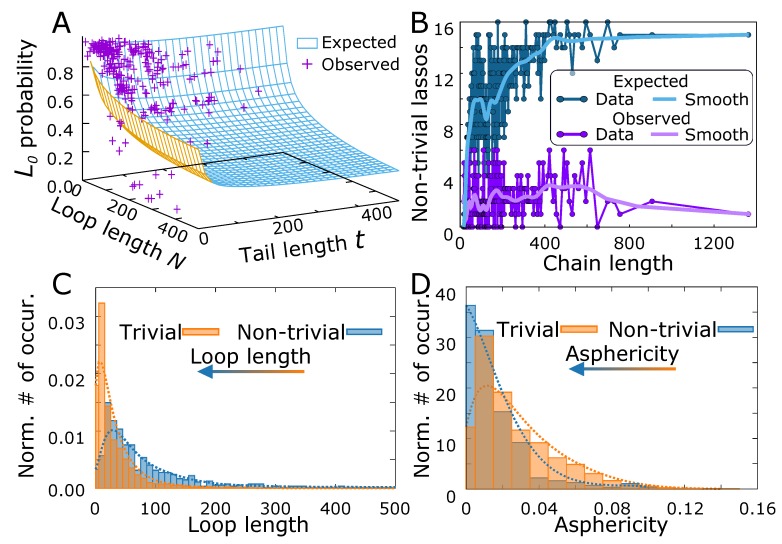
The complex lasso shapes in proteins. (**A**) The probability of trivial protein lasso versus the trivial lasso polymer surface, as a function of loop and tail length. (**B**) The expected and observed number of the non-trivial lassos as a function of the chain length. The light color curves show the smoothed traces. The normalized histogram of (**C**) loop lengths and (**D**) asphericity for the pierced and non-threaded (trivial) case. The arrows denote the shift of the distribution from trivial to the non-trivial case. The dotted lines denote the smoothed fit.

**Table 1 polymers-11-00707-t001:** Parameters fitted for the shape parameters scaling as the function of the thread thickness. The “ratio” is the value of the preexponential factor compared with the case of unthreaded loop $a_{r}\left(thickness=n\right)/a_{r}\left(thickness=0\right)$. The scaling factor for the asphericity for the unthreaded loop was given in italic as it stands out from the trend and the fitting error obtained in this case was significant (see Appendix A).

Thread	Radius of Gyration			Asphericity			Prolatness	
Thickness	$a_{R}\left[\mathrm{nm}\right]$	$c_{R}\left[\mathrm{nm}\right]$	ratio	$a_{A}$	μ	$A_{\infty }$	$a_{P}$	$P_{\infty }$
0	0.154	−0.102	1.00	0.0342	*−0.186*	0.0708	−1.37	0.300
1	0.164	−0.0825	1.06	0.188	−0.753	0.0853	−3.07	0.315
2	0.164	−0.0162	1.06	0.353	−0.822	0.0858	−4.57	0.384
3	0.164	0.0376	1.06	0.845	−0.993	0.0869	−5.53	0.420
4	0.164	0.106	1.06	1.05	−0.969	0.0856	−6.10	0.413

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
