# Peer review of "Statistical Properties of Lasso-Shape Polymers and Their Implications for Complex Lasso Proteins Function"

_polymers, 2019, doi:10.3390/polym11040707_

Reviewer 1 Report

Proteins displaying non-trivial topologies are a fascinating subject that offers ever new surprises. Whether these topologies are rare, or common because of some robust physical principle, and what is their biological function if any, are very interesting questions. In this manuscript, the authors investigate some features of complex lasso topologies with the aim of advancing the understanding of complex lasso-proteins. In particular, the authors observe that non-trivial lasso proteins are rather common, and compare their occurrence in polymers. The manuscript reports research that is well conducted and of potential interest for a specialized community. I, therefore, recommend the publication.

On a minor note, the English could be improved in several points. In particular, some sentences sound a bit childish (eg "he polymer theory was coined in the past century by great scientists"), or when the authors talk about "statistical probability".

Also, in the discussion, the authors could write and comment more clearly whether lasso-proteins are more common than expected than not, which is one of the aims of the paper, and which in the current form is not so clearly discussed.

A list of minor points:

4 and 45 phantom and trivial lasso loop -> never defined what they mean

5 strictly positive -> non-zero

10, 11 rather childish

41 not clear what least-polymeric means here

Figure 2 “Statistical probability” is redundant

Figure 3 Figures captions should stand on their own, please explain what the domination number is. Also in the labels, what is "other"?

Author Response

Proteins displaying non-trivial topologies are a fascinating subject that offers ever new surprises. Whether these topologies are rare, or common because of some robust physical principle, and what is their biological function if any, are very interesting questions. In this manuscript, the authors investigate some features of complex lasso topologies with the aim of advancing the understanding of complex lasso-proteins. In particular, the authors observe that non-trivial lasso proteins are rather common, and compare their occurrence in polymers. The manuscript reports research that is well conducted and of potential interest for a specialized community. I, therefore, recommend the publication.

On a minor note, the English could be improved in several points. In particular, some sentences sound a bit childish (eg ”the polymer theory was coined in the past century by great scientists”), or when the authors talk about ”statistical probability”.

We thank the reviewer for positive comments on our work. To improve our writing, we asked for help English native speakers and introduced the corrections. Next, we again proofread the manuscript and checked the typos with the on-line tools.

Concerning the “childish” expressions, it is the result of the balance we want to keep between scientific rigorous and the manuscript being understandable for the broad community. Still, we corrected the mentioned sentence. We also shorted “statistical probability” to “probability” throughout the manuscript, although in the methods we clarify, that we calculate the fraction of the numbers of observed structures, called by us the “statistical probability”, as it is only an approximation to a real probability distribution.

Also, in the discussion, the authors could write and comment more clearly whether

lasso-proteins are more common than expected than not, which is one of the aims of the

paper, and which in the current form is not so clearly discussed.

We are sincerely grateful for this reviewer’s insight! In fact, that was not the original aim of the work, although some expressions could suggest otherwise. However, this definitely lies in the scope of the manuscript. Therefore, we decided to enhance the work by more detailed analysis of the referee’s comment. In particular, we added two new paragraphs and modified the last figure, inserting two new panels, showing two approaches to comparison of numbers of polymeric and biological non-trivial lassos.

A list of minor points:

·    4 and 45 phantom and trivial lasso loop -> never defined what they mean

·    5 strictly positive -> non-zero

·      10, 11 rather childish

·      41 not clear what least-polymeric means here

·      Figure 2 “Statistical probability” is redundant

·      Figure 3 Figures captions should stand on their own, please explain what the

·      domination number is. Also in the labels, what is ”other”?

We are grateful for those specific comments. We’ve corrected all of the suggested fragments.

Reviewer 2 Report

The manuscript addresses the study of statistical properties of so-called lassos, polymer motifs characterised by a closed loop threaded one or more times by a tail.

The authors study the probability of formation of these lasso structures by means of equilateral random walk generation, outlining the dependence of this probability on the size of loop and tail.

Moreover a study on the effect of lasso structure on the shape of the polymeric loop is obtained with a coarse grained model of the threaded ring.

Finally, the statistical behaviour of these simple models is compared to a set of complex lasso proteins, to identify possible functional candidates, and support the hypothesis that lasso structure might be functional.

The article focuses on an interesting problem, providing, in my opinion, a moderate but useful advancement to the field.

My main concerns about the paper regard the form and exposition. I have had difficulties in understanding the methodologies used, and I have to admit that, even after reading it a few times, there are still things that remain unclear.

First, the authors should explain better how the phantom loop is generated. For example I don't understand if the tail is connected to the loop or not.

The term "phantom loop" is also a bit specific, and the authors should define its meaning. Regarding this, I think it would help to provide further explanation also to the S1.1/2/3 sections in the supplementary information.

Also the methodological part about the coarse-grained model is obscure to me. The authors should say explicitly that there is excluded volume, even though the result make it clear later (when they write that long-distance contacts are removed this can in principle include also repulsive interaction). In this sense Fig. 4A would be more helpful in the methodological part of the paper.

The discussion of the results is, in general, more clear despite some statements are not completely supported by data or literature (for example lines 163-165).

I also think that the results on the selection of possible functional lasso structures should be provided in a more quantitative way, for example providing average probabilities of lasso formation for the different classes of proteins mentioned in the paragraph of line 274, or by providing the spectrum of lasso probabilities.

Finally, there are many typos and the language form should be improved substantially.

Author Response

Reviewer’s #2 comments:

The manuscript addresses the study of statistical properties of so-called lassos, polymer motifs characterised by a closed loop threaded one or more times by a tail. The authors study the probability of formation of these lasso structures by means of equilateral random walk generation, outlining the dependence of this probability on the size of loop and tail. Moreover a study on the effect of lasso structure on the shape of the polymeric loop is obtained with a coarse grained model of the threaded ring. Finally, the statistical behaviour of these simple models is compared to a set of complex lasso proteins, to identify possible functional  candidates, and support the hypothesis that lasso structure might be functional. The article  focuses on an interesting problem, providing, in my opinion, a moderate but useful  advancement to the field.

My main concerns about the paper regard the form and exposition. I have had difficulties in understanding the methodologies used, and I have to admit that, even after reading it a few times, there are still things that remain unclear. First, the authors should explain better how the phantom loop is generated. For example I don’t understand if the tail is connected to the loop or not.

The term ”phantom loop” is also a bit specific, and the authors should define its meaning. Regarding this, I think it would help to provide further explanation also to the S1.1/2/3 sections in the supplementary information.

Also the methodological part about the coarse-grained model is obscure to me. The authors should say explicitly that there is excluded volume, even though the result make it clear later (when they write that long-distance contacts are removed this can in principle include also repulsive interaction). In this sense Fig. 4A would be more helpful in the methodological part of the paper.

We thank the reviewer for positive comments on our work. The possibly insufficient description in the methodological part was the result of our attempts to keep the manuscript concise. However, following the referee’s comment, we have extended our methodology part and took into the consideration all particular remarks.

The discussion of the results is, in general, more clear despite some statements are not completely supported by data or literature (for example lines 163-165).

The particular statement was our observation, leading us to the final model used. However, this do not matter much for the results, therefore, in order to keep the work concise, we removed the questioned sentence.

I also think that the results on the selection of possible functional lasso structures should be provided in a more quantitative way, for example providing average probabilities of lasso formation for the different classes of proteins mentioned in the paragraph of line 274, or by providing the spectrum of lasso probabilities.

We thank the reviewer for pointing this out. We thought that the raw numbers, including the calculated probabilities present in the Supplementary Information are sufficient, therefore we did not include more quantitative description in the main text. However, following the referee’s remark, we added the numbers of proteins in each class mentioned in the paragraph with respective average probabilities. Moreover, we plotted the spectrum of lasso probabilities, which we included in SI. Furthermore, in the table with the data in SI, we singled out the miniproteins and all the proteins subjected to the analysis of the function of lasso motif. 

Finally, there are many typos and the language form should be improved substantially.

We thank the reviewer for his alertness. Again as in the case of the first reviewer, we have made every effort to improve the linguistic aspect of our article.

Round  2

Reviewer 2 Report

The presentation of the manuscript has been improved, and many typos and errors have been removed. 

I list some minor points/typos in the following:

1 - Abstract : phantom lasso is mentioned without being defined

2 - line 29 : some motifs ARE

3 - line 43 : specify the polymer model you use for the prediction. e.g. "from the structure generated via random walk".

4 - line 48 : The definition of phantom lasso is still not clear. How is the tail connected to the loop? Phantom lassos are not just "polymers deprived ..." they are composed by a loop and a tail.

5 - line 60 : I would refer to Fig 2A

6 - line 75 : AT temperature T

7 - line 161 : p was already defined before the equation

8 - line 175 : refer to eq. 6, defining the distention

9 - line 204 : the thread thickness does not influence the value of Rg asymptotically

10 - line 227 : non-threaded

11 - line 265 : ONE OF the biological realizations...

12 - line 270 : versus the probability of trivial lassos occurrence in phantom lasso polymers.

13 - line 277 : The procedure is still not clear to me. Maybe for clarity it would be beneficial to add few words to specify the procedure (even though it is explained in detail in the SI).

S1 - "analogical" is not probably the correct word

S8 - I don't understand the "number of loops" on the y axis of the spectrum.

S8 - EncompaSS

Author Response

We thank the scrupulous reviewer, who was able to list some additional points, despite our meticulous analysis of the manuscript. We included all the referee’s remarks, listed below (in blue).

Reviewer’s comments:

The presentation of the manuscript has been improved, and many typos and errors have been removed. 

I list some minor points/typos in the following:

1 - Abstract : phantom lasso is mentioned without being defined

2 - line 29 : some motifs ARE

3 - line 43 : specify the polymer model you use for the prediction. e.g. "from the structure generated via random walk".

4 - line 48 : The definition of phantom lasso is still not clear. How is the tail connected to the loop? Phantom lassos are not just "polymers deprived ..." they are composed by a loop and a tail.

5 - line 60 : I would refer to Fig 2A

6 - line 75 : AT temperature T

7 - line 161 : p was already defined before the equation

8 - line 175 : refer to eq. 6, defining the distention

9 - line 204 : the thread thickness does not influence the value of Rg asymptotically

10 - line 227 : non-threaded

11 - line 265 : ONE OF the biological realizations...

12 - line 270 : versus the probability of trivial lassos occurrence in phantom lasso polymers.

13 - line 277 : The procedure is still not clear to me. Maybe for clarity it would be beneficial to add few words to specify the procedure (even though it is explained in detail in the SI).

S1 - "analogical" is not probably the correct word

S8 - I don't understand the "number of loops" on the y axis of the spectrum.

S8 – EncompaSS

As stated above, we included all the reviewer’s remarks. In particular, we expanded the procedure used to obtain Fig. 6B (remark 13), and rephrased the point-by-point explanation in the SI. We hope, that now the procedure is crystal-clear.
